

# Changes in soil carbon, nitrogen, and phosphorus in *Pinus massoniana* forest along altitudinal gradients of subtropical karst mountains

Kun Nie, Ming Xu and Jian Zhang

Key laboratory of Plant Resource Conservation and Germplasm Innovation in Mountainous Region (Ministry of Education), Collaborative Innovation Center for Mountain Ecology & Agro-Bioengineering (CICMEAB), College of Life Sciences, Guizhou University, Guiyang, Guizhou Province, China

## ABSTRACT

Changes in altitude have a long-term and profound impact on mountain forest ecosystems. However, there have been few reports on changes in soil carbon, nitrogen, and phosphorus contents (SCNPC) along altitudinal gradients in subtropical karst mountain forests, as well as on the factors influencing such changes. We selected five *Pinus massoniana* forests with an altitudinal gradient in the karst mountain area of Southwest China as research objects and analyzed the changes in SCNPC along the altitudinal gradient, as well as the influencing factors behind these changes. Soil organic carbon, total nitrogen, and available nitrogen contents first increased and then decreased with increasing altitude, whereas the contents of total phosphorus and available phosphorus showed no obvious trend. In the karst mountain *P. massoniana* forest, SCNPC in the topsoil is most significantly affected by total glomalin-related soil protein (TG) and soil moisture content (SMC) (cumulative explanatory rate was 45.28–77.33%), indicating that TG and SMC are important factors that affect SCNPC in the karst mountain *P. massoniana* forest. In addition, the main environmental factors that affect SCNPC in the subsoil showed significant differences. These results may provide a better scientific reference for the sustainable management of the subtropical mountain *P. massoniana* forest.

# INTRODUCTION

The effects of altitudinal changes on the biodiversity and environmental factors of mountain ecosystems are a hot topic in ecological research (*Körner, 2007*; *Zhang et al., 2022a*; *Zhang et al., 2022b*). Altitudinal gradients have been used as "natural experiments" to understand the influence of temperature, light, rainfall, and other environmental factors on soil physicochemical properties in mountain ecosystems (*Maja, Ranko & Bishal, 2011*; *Wu et al., 2013*). In particular, altitude is one of the most important factors that determines micro-climatic conditions (temperature and precipitation) that impact plant/microbe distribution and growth (*Pan et al., 2009*; *Zhang et al., 2022a*; *Zhang et al., 2022b*), which eventually impacts soil carbon, nitrogen, and phosphorus contents (SCNPC) in mountain

Corresponding author
Jian Zhang, zhangjian12102@163.com

forests (*Körner, 2007*). It is of great importance to understand the changes in and factors influencing SCNPC in mountain forest soils at different altitudes to reveal the interaction between biogeochemical cycles and altitudinal gradients (*Njeru et al., 2017*; *Zhang et al., 2019a*; *Zhang et al., 2019b*). However, the trend of SCNPC in mountain forests along altitudinal gradient remains debatable. Previous studies have shown that with an increase in altitude, SCNPC in forests can show an increasing (*He et al., 2022*; *He et al., 2021a*; *He et al., 2021b*; *Qiu, Lin & Wang, 2022*), unimodal (*Chen et al., 2010*; *Li et al., 2017*; *Yang et al., 2018*), or decreasing trend (*Bhandari & Zhang, 2019*; *Maja, Ranko & Bishal, 2011*; *Wilcke et al., 2008*) or have no correlation (*Tan & Wang, 2016*; *Zhu et al., 2022*). Thus, the variation in SCNPC in mountain forests with an altitudinal gradient can greatly differ or be quite uncertain prior to the analysis of the given forest. Moreover, the variation in SCNPC in mountain forests is often affected by natural and anthropogenic factors, such as soil physicochemical properties (*e.g.*, soil parent material (*Pichler et al., 2021*), soil texture (*Giardina et al., 2001*), soil bulk density (*Manrique & Jones, 1991*), soil pH (*Zhou et al., 2019*), and glomalin-related soil protein (*Li et al., 2022*), topographic factors (*e.g.*, altitude (*He et al., 2016*) and slope (*Bangroo, Najar & Rasool, 2017*), climatic factors (*e.g.*, temperature (*Tashi et al., 2016*) and rainfall (*Austin & Sala, 2002*), biological factors (*e.g.*, plant *De Feudis et al., 2016*), animal (*Haimi, 2000*), and microbial communities (*Nottingham et al., 2019*), and disturbance factors (*e.g.*, forest management (*Karla & Leopoldo, 2021*), forest fire (*Certini, 2005*), and human activity (*Ahmedin & Elias, 2022*; *Zhao et al., 2015*)). Therefore, strengthening our knowledge of the distribution of soil carbon, nitrogen, and phosphorus along altitudinal gradients as well as the mechanism influencing relevant trends will be helpful for achieving a better understanding and scientific management of mountain forest ecosystems (*Bangroo, Najar & Rasool, 2017*; *Feng, Bao & Pang, 2017*; *Njeru et al., 2017*).

The karst mountains in Southwest China have large topographical differences, habitat fragmentation, high spatial heterogeneity, and a fragile eco-environmental system (*He et al., 2022*; *Yuan, 2006*; *Liu et al., 2022a*; *Liu et al., 2022b*; *Zhang et al., 2012*), and in addition, the soil in karst areas are low in quantity with, low rate of parent rock soil formation, high content of calcium and magnesium, high exposure rate of bedrock, and high risk of underground leakage (*Smith et al., 2013*; *Bai et al., 2020*). Improvement and rational utilization of karst soils are important for soil sustainability and productivity, and hance, vegetation restoration has become an important measure to prevent stone desertification and promote ecological sustainability in karst mountains (*Zhong et al., 2022*). *Pinus massoniana*, as a pioneer tree species that is currently involved in the vegetation succession in this region, has rapid organic matter accumulation and superb adaptability to harsh environments. The *P. massoniana* forest constitutes the main forest vegetation type in southern China and plays an important role in the karst mountain forest ecosystem of this region (*Huang et al., 2020*; *Lei et al., 2018*; *Zhang et al., 2022a*; *Zhang et al., 2022b*). Research on the characteristics of vertical variation of soil nutrients and their influencing factors in subtropical mountain horsetail forests with different altitudinal gradients is important to enhance the scientific management of regional forests in this region. The widely distributed *P. massoniana* forest in Southwest China provides a natural experimental

object that is well-suited for studying the changes in SCNPC in a single forest type along an altitudinal gradient. Therefore, this study selected *P. massoniana* forests along an altitudinal gradient (1,200-1,600 m a.s.l.) in the mountains of central Guizhou as the research object, with the specific aims to: (1) reveal the changes of SCNPC along an altitudinal gradient in the *P. massoniana* forest of a karst mountain ecosystem; (2) analyze the main factors influencing differences in SCNPC; and (3) explore the quantitative relationship between SCNPC and its main influencing factors. This study will improve our understanding of soil nutrient cycling and major drivers in fragile karst forest ecosystems, and provide a theoretical basis for forest management especially in terms of forest soil carbon neutrality in the context of global climate change.

## MATERIALS AND METHODS

### Study area

Longli County (26°21′~26°41′E, 106°51′~107°11′N), located in the center of Guizhou Province, is a mountainous area in southern China with karst topography and an altitude of 765–1,766 m a.s.l. (Fig. 1). The terrestrial vegetation in this region mostly consists of evergreen broad-leaved forest. And the coniferous and broad-leaved mixed forest mostly grows in less-disturbed areas with limestone soil (*Li et al., 2008*). These forests mainly include *P. massoniana*, and secondary *Quercus fabri*, and *Cunninghamia lanceolata* forests. The main climate type is a subtropical monsoon humid climate, with a mean annual precipitation of approximately 1,100 mm, a mean annual temperature of 15.0 °C, a mean temperature of the coldest month (January) of 4.8 °C, a mean temperature of the hottest month (July) of 23.5 °C, and a mean annual sunshine duration of approximately 1,160 h (*Luo et al., 2019*).

### Plot design and sampling

Five altitudes were selected every 100 m from 1,200 m a.s.l. to 1,600 m a.s.l. at the Longli Forest Farm. At each altitude, three independent replicate sampling plots (20 m × 20 m) of relatively consistent stand age were created as replicates. The development time of the *P. massoniana* forest at 1,200–1,500 m a.s.l. was the same, and the average age of the *P. massoniana* forest was 40–53 a. However, owing to the limited distribution of extant *P. massoniana* forests in the 1,600-m altitudinal gradient range of the study area (close to the upper altitude limit of the natural distribution of subtropical *P. massoniana*), the stand age selected for this study consisted mainly of young *P. massoniana* of approximately 10 a (Fig. 1). Basic environmental information for each sampling plot was recorded, and a vegetation survey was carried out.

From each sampling plot, soil samples were taken from six random points using a soil drill (diameter, 4.5-cm), to collect the topsoil (0–20 cm) and subsoil (20–40 cm). Soil samples from the same depth were consolidated and thoroughly mixed into one sample per plot. Each consolidated sample was air-dried, ground, and sieved through a two mm mesh sieve to measure its basic physicochemical properties. Additionally, three soil core samples (0–20 cm) were collected from each sampling plot to determine the soil bulk density (SBD) and soil moisture content (SMC).

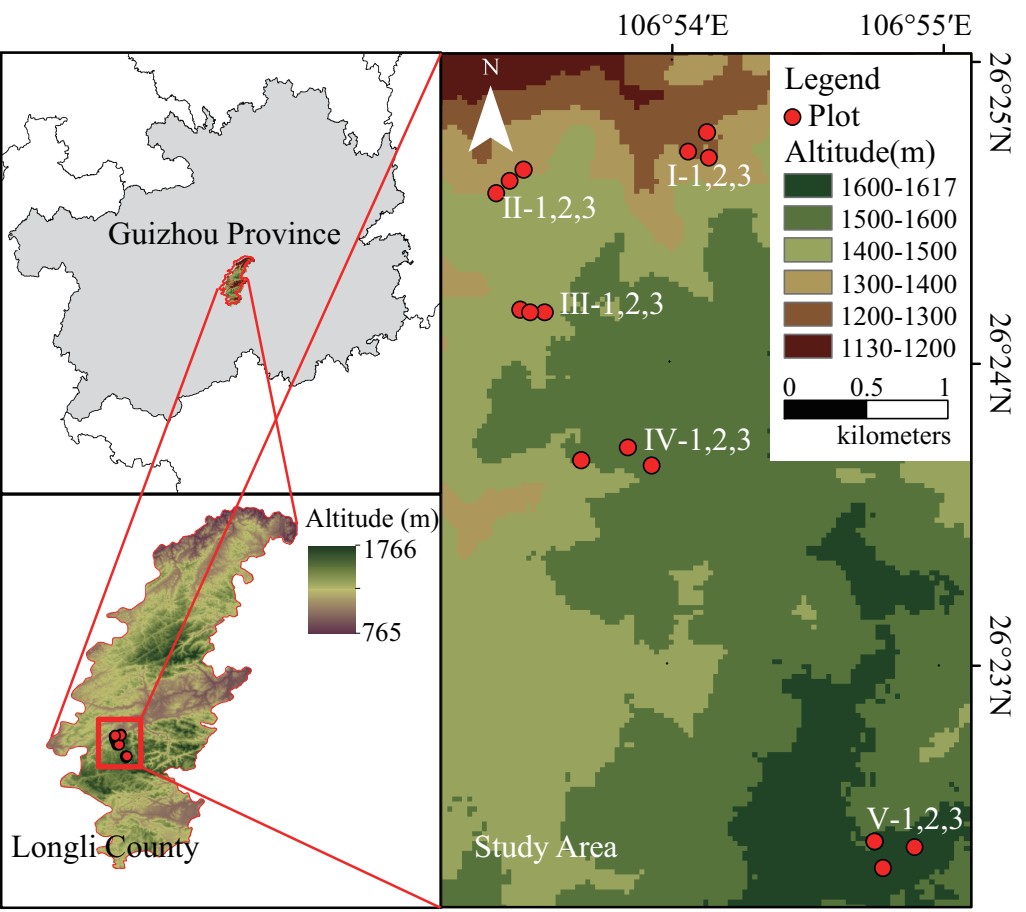

**Figure 1** Location of the study area and the sampling plot.

## Soil physicochemical proprieties analysis

Soil organic carbon content (SOC) was obtained using the $K_2Cr_2O_7$-$H_2SO_4$ oxidation method (*ISSCAS, 1978*). Soil total nitrogen content (TN) was measured using the Kjeldahl digestion procedure, and soil available nitrogen content (AN) was determined using the alkaline hydrolysis diffusion method (*ISSCAS, 1978*). Soil total phosphorus content (TP) was measured using a UV-2600 spectrophotometer (Shimadzu, Kyoto, Japan) after $H_2SO_4$–$HClO_4$ digestion, and soil available phosphorus content (AP) was determined using the HCl-$H_2SO_4$ extraction solution colorimetric method (*ISSCAS, 1978*). Soil exchangeable calcium (eCa) was measured using Shimadzu AA7000 atomic absorption spectrophotometer (Shimadzu, Kyoto, Japan) after $NH_4OAc$ extraction (*ISSCAS, 1978*). The soil pH was measured in the suspension (soil/water = 1:2.5) using a pH meter (*Sheng et al., 2021*). After the soil was dried in an oven for 12 h at 105 °C, the SBD was determined as the dry weight per unit volume of soil core, and SMC was determined gravimetrically by weighing. All the 28 soil samples were analyzed using the laser diffraction method for their soil particle size distribution using a Bettersizer 2,600 laser particle size analyzer (Bettersize Instruments Ltd, Liaoning, China), and the percentages of clay (<0.002 mm), silt

(0.002–0.05 mm) and sand (0.05–2 mm) were calculated according to the US soil taxonomy classification system (*Zhang et al., 2022a*). The pre-treatment was done by adding 10 mL of 10% hydrogen peroxide ($H_2O_2$) and 10% hydrochloric acid (HCl) to remove organic matter and carbonates. In addition, 10 mL of Calgon ($(NaPO_3)_6$) with a concentration of 0.05 mol L $^{-1}$ was added and then stirred to fully separate the primary soil particles. Before laser diffraction analysis, ultrasonic dispersion for 30 min was applied to samples. Based on the differences in the availability and turnaround time of glomalin-related soil proteins (GRSP) in soil, they can be divided into easily extractable glomalin-related soil proteins (EEG) and total glomalin-related soil proteins (TG). In this study, EEG and TG were determined separately using a modified Kormas Brilliant Blue colorimetry method (*Gao, Peng & Wu, 2019*; *Li et al., 2022*).

### Statistical analysis

All statistical analyses and mapping were conducted using R 4.2.1 software. The Kruskal–Wallis *H* test (KWH) was employed to identify the differences in SCNPC in the *P. massoniana* forest along the altitudinal gradient. If significant effects ($P < 0.05$) were observed by KWH, comparison among the means was performed using Dunn's test. The Mann–Whitney *U* test was used to identify the differences in SCNPC between the topsoil and subsoil in the *P. massoniana* forest. To investigate the variation in SCNPC along the altitudinal gradient, the curve-fitting regressions of SOC, TN, AN, TP, and AP with altitude were analyzed using the ggtrendline package. Spearman's correlation analysis was performed to identify correlations between several environmental variables (*e.g.*, altitude, slope, pH, SMC, SBD, clay, silt, sand, TG, and EEG) and SOC, TN, AN, TP, and AP. Correlations among these variables were also assessed. When more than one variable was available within a component, it was necessary to consider the multicollinearity caused by the significant correlation between environmental variables (*Jagadamma et al., 2008*). The relationships between SOC, TN, AN, TP, or AP and environmental variables were then obtained using multiple regression analysis according to the stepwise method, using environmental variables as independent variables and SOC, TN, AN, TP, and AP as dependent variables (*Maja, Ranko & Bishal, 2011*; *Wang, Wang & Ouyang, 2012*). The rdacca.hp package contains two main functions: rdacca.hp and permu.hp. The former conducts both variation and hierarchical partitioning for canonical analysis without limiting the number of predictors (or matrices of predictors), whereas the latter implements significance testing for individual predictor (or a matrix of predictors) contribution using a permutation test (*Lai et al., 2022*). Therefore, it was used to calculate the relative explanatory rate of each environmental factor retained in the stepwise multiple regression model to the explainable part of the total variance in SCNPC (*Kong et al., 2022*; *Lai et al., 2022*).

## RESULTS

### Changes in SCNPC in the *P. massoniana* forest at different altitudes

In the *P. massoniana* forest of the karst mountains in the study area, the SCNPC significantly differed at the five altitudinal gradients. The SOC, TN, AN, and TP of the topsoil and subsoil

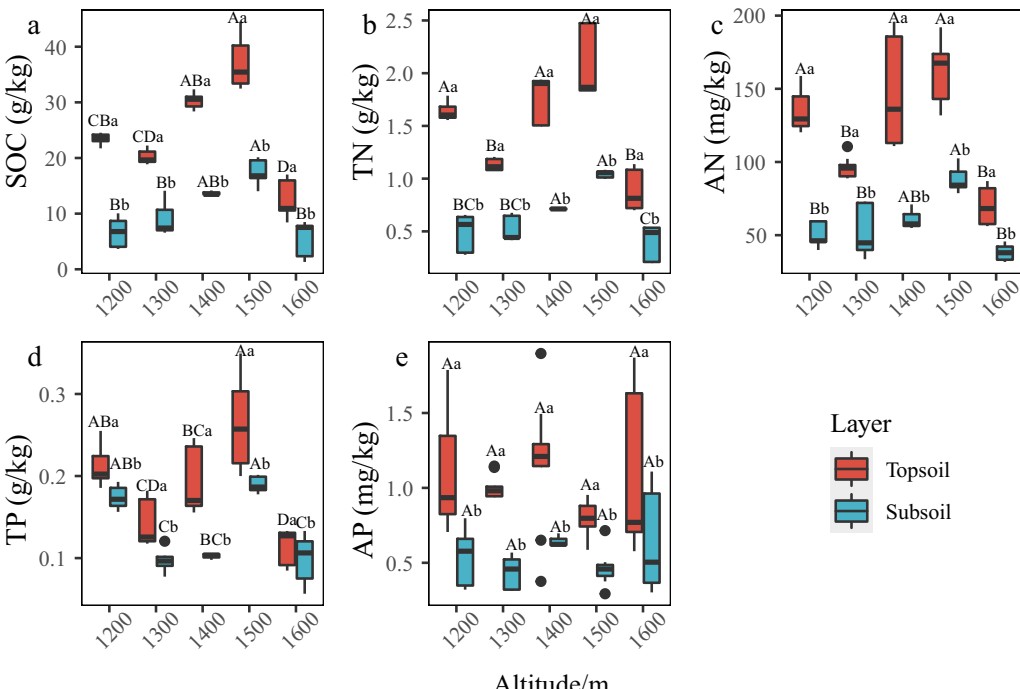

**Figure 2 Characteristics of soil carbon, nitrogen and phosphorus content (SCNPC) in the *Pinus massoniana* forest at different altitudes.** SOC, soil organic carbon; TN, soil total nitrogen content; AN, soil available nitrogen content; TP, soil total phosphorus content; AP, soil available nitrogen content; topsoil, 0–20 cm soil; subsoil: 20–40 cm soil. Different capital letters above the box indicate significant differences between altitude at the same soil layer, and different lowercase letters above the box indicate significant differences between soil layer during the same altitude ($p < 0.05$).

at 1,500 m a.s.l. were significantly higher than those at other altitudes ($P < 0.05$), but there was no significant difference in AP among the five altitudes ($P > 0.05$, Fig. 2). The fitted results of SCNPC with altitude showed that the SOC, TN, and AN in the topsoil and subsoil first increased and then decreased with increasing altitude (Figs. 3A–3C, $P < 0.05$), whereas TP and AP did not significantly change (Figs. 3D & 3E). In addition, compared to the SCNPC in the topsoil, that in the subsoil was significantly reduced, showing clear surface accumulation phenomenon ($P < 0.05$, Fig. 2).

## Correlation between SCNPC and environmental factors in the *P. massoniana* forest

There was a significant correlation between SCNPC and environmental factors at different altitudes in the *P. massoniana* forest. Among them, the SOC, TN, AN, and TP of the topsoil in the *P. massoniana* forest had a significantly positive correlation with eCa, SMC, clay, silt, EEG, and TG ($R > 0.272$), and a significantly negative correlation with SBD and sand ($R < -0.396$, $P < 0.05$, Fig. 4). The SOC, TN, AN, and TP of the subsoil in the *P. massoniana* forest were positively correlated with SMC, EEG, and TG ($R > 0.339$), and negatively correlated with SBD ($R < -0.422$, $P > 0.05$). The AP of the *P. massoniana* forest was negatively correlated with the SMC of the topsoil ($R = -0.104$) and the TG of both

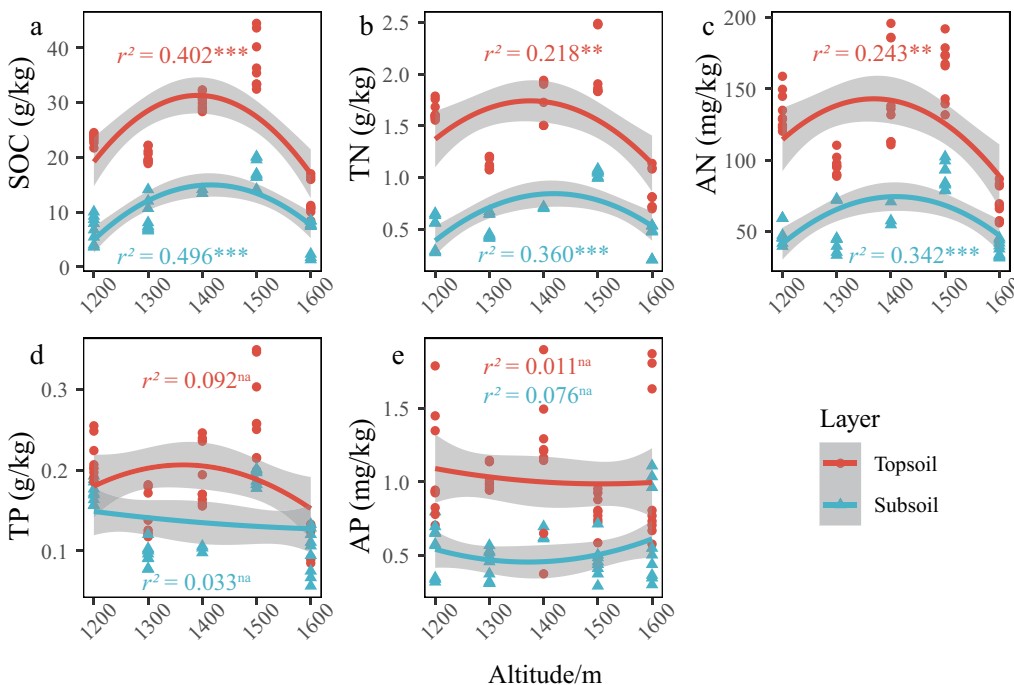

**Figure 3** **Changes in the SCNPC along the altitudinal gradient in *Pinus massoniana* forests.** SOC, soil organic carbon; TN, soil total nitrogen content; AN, soil available nitrogen content; TP, soil total phosphorus content; AP, soil available nitrogen content; topsoil, 0–20 cm soil; subsoil, 20–40 cm soil. $r^2$, adjustment r squared of the regression equation; na, $p$ value of the regression equation ($P$) is greater than 0.05; *, $P$ is less than 0.05; **, $P$ is less than 0.01; ***, $P$ is less than 0.001.

the topsoil ($R = -0.122$) and subsoil ($R = -0.042$). In addition, there are also significant correlations among various environmental factors, such as altitude (Alt) and slope (Slp), pH, Silt, TG, *etc.*

## Relationships between SCNPC and environmental factors in the *P. massoniana* forest based on stepwise multiple regression analysis

The ten stepwise multiple regression equations, between SCNPC and environmental factors in the topsoil and subsoil of the *P. massoniana* forest, were significant ($P < 0.05$, Table 1). The main influencing factors of SCNPC differed significantly among different soil layers (Fig. 5). In the topsoil, the environmental variables retained in the five regression equations jointly explained 86.56, 77.59, 67.96, 73.67, and 15.95% of the vertical variability of SOC, TN, AN, TP, and AP contents vertical variability, respectively (Table 1). Among them, the cumulative explanatory rates of TG, SMC, Silt, SBD, and Alt to the explainable part in the SOC, TN, AN, TP, or AP of the topsoil were all greater than 88.72%. The cumulative explanatory rates of TG and Alt to AP of topsoil were equal to 93.25%. The cumulative explanatory rates of TG and SMC to the explainable part of SOC, TN, AN, TP and AP in the topsoil amounted to 45.28–77.33% (Fig. 5). These results indicate that TG, soil physical properties (SMC, Silt, and SBD), and Alt may be important factors influencing the variation in SCNPC of topsoil in the *P. massoniana* forest of karst mountains along an
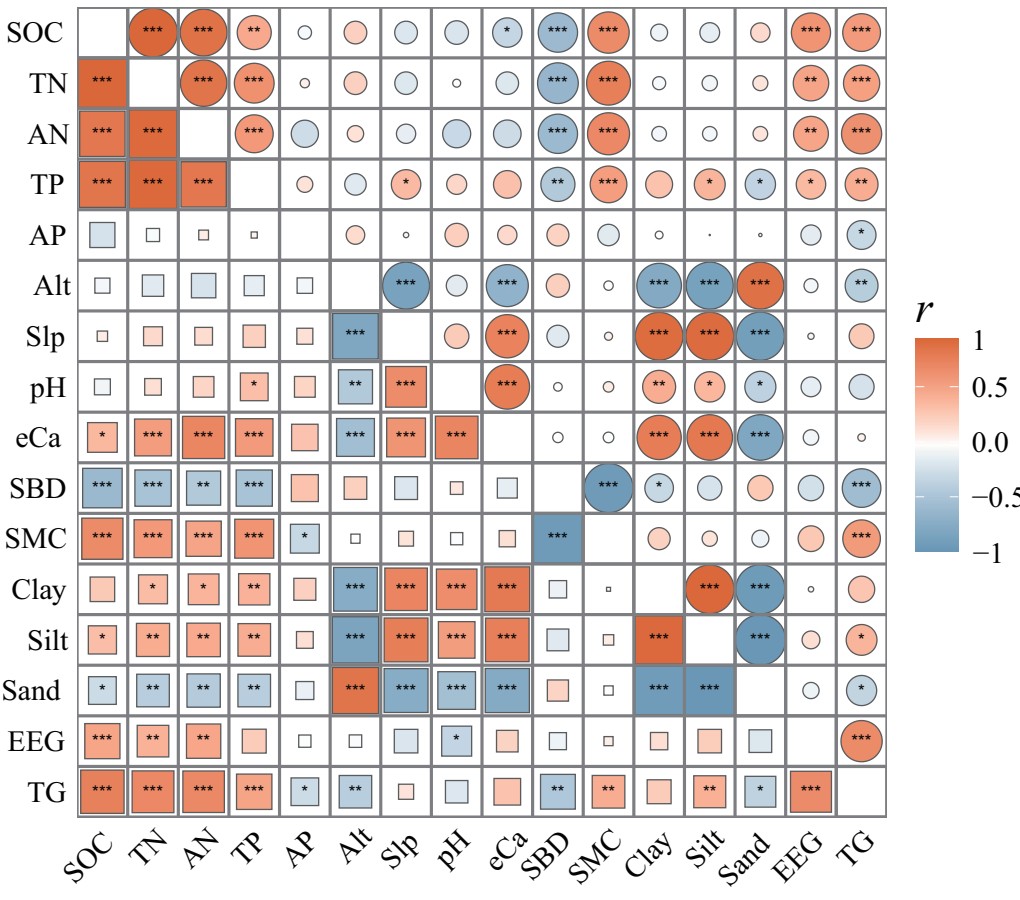

**Figure 4 Correlation matrix of SCNPC with environmental factors in *Pinus massoniana* forests.** SOC, soil organic carbon; TN, soil total nitrogen content; AN, soil available nitrogen content; TP, soil total phosphorus content; AP, soil available nitrogen content; Alt, altitude; Slp, slope; eCa, soil exchangeable calcium; SBD, soil bulk density; SMC, soil moisture content; Clay, percentages of soil clay (< 0.002 mm); Silt: percentages of soil silt (0.002–0.05 mm); Sand: percentages of soil sand (> 0.05 mm); EEG, easily extractable glomalin-related soil protein; TG, total glomalin-related soil protein. The different colored squares in the lower left corner represent the topsoil (0–20 cm), and the different colored circles in the upper right corner represent the subsoil (20–40 cm). The size of circle and squares shows the size of $r$-value between two indexes. The closer the r-value is to 0, the smaller the circle or square is, and the closer it is to 1 or −1, the larger it is. *, $P < 0.05$; **, $P < 0.01$; ***, $P < 0.001$.

altitudinal gradient, with TG and SMC being the main influencing factors. In the subsoil, the environmental variables retained in the five regression equations jointly explained 68.82, 69.72, 64.52, 72.63, and 8.06% of the vertical variability SOC, TN, AN, TP, and AP contents, respectively (Table 1). Among them, the cumulative explanatory rates of SMC and EEG to the explainable part of SOC and TN was greater than 92.95%. The cumulative explanatory rates of SMC and TG to the AN was 86.89%. The cumulative explanatory rates of SMC (>25.99%), Slp, silt, SBD, and clay to TP was greater than 10%, while AP was only explained by the TG (Fig. 5). These results indicate that there are significant differences in the main influencing factors of subsoil C, N, and P, where SMC, TG, and EEG are

**Table 1 Stepwise multiple regression equation of SCNPC versus environmental factors in the topsoil and subsoil in the *P. massoniana* forest.**

| Soil layers | Stepwise multiple regression equation | $R^2$ | $P$ |
|---|---|---|---|
| topsoil | $SOC = -141.80 + 55.06TG + 0.052Alt + 0.36SMC -4.01pH + 18.52SBD + 0.12Silt + 0.73Clay + 0.24Slp$ | 0.8656 | <0.001 |
| topsoil | $TN = -8.36 + 2.59TG + 0.0023Alt + 0.017SMC + 0.28pH + 0.83SBD + 0.016Silt$ | 0.7759 | <0.001 |
| topsoil | $AN = -653.40 + 245.42TG + 0.18Alt + 0.42SMC + 40.83pH + 0.97Silt$ | 0.6796 | <0.001 |
| topsoil | $TP = -1.22 + 0.23TG + 0.0003Alt + 0.0031SMC + 0.065pH + 0.16SBD + 0.0021Silt$ | 0.7367 | <0.001 |
| topsoil | $AP = 3.66 - 2.14TG - 0.0009Alt + 1.1EEG$ | 0.1595 | = 0.017 |
| subsoil | $SOC = -20.52 + 0.15SMC + 62.56EEG -1.88pH + 0.0077Alt$ | 0.6882 | <0.001 |
| subsoil | $TN = -1.32 + 0.0093SMC + 0.00044Alt + 2.46EEG$ | 0.6972 | <0.001 |
| subsoil | $AN = -56.83 + 0.49SMC + 0.039Alt + 159.85TG -9.59pH$ | 0.6452 | <0.001 |
| subsoil | $TP = -0.82 + 0.0031SMC + 0.00025Alt + 0.15TG + 0.0057Slp + 0.028pH + 0.16SBD -0.013Clay + 0.0025Silt$ | 0.7263 | <0.001 |
| subsoil | $AP = 1.06 - 1.07TG$ | 0.0806 | = 0.044 |

Notes.

topsoil, 0–20 cm soil; subsoil, 20–40 cm soil; SOC, soil organic carbon; TN, soil total nitrogen content; AN, soil available nitrogen content; TP, soil total phosphorus content; AP, soil available nitrogen content; Alt, altitude; Slp, slope; SBD, soil bulk density; SMC, soil moisture content; Clay, percentages of soil clay (< 0.002 mm); Silt, percentages of soil silt (0.002–0.05 mm); EEG, easily extractable glomalin-related soil protein; TG, total glomalinrelated soil protein; R2, coefficient of determination; P, *p* value.

important factors that affect SCNPC of subsoil in *P. massoniana* forests of karst mountains along an altitudinal gradient (Fig. 5).

## DISCUSSION

### Changes in SCNPC of the *P. massoniana* forest at different altitudes

In the karst mountain *P. massoniana* forests in central Guizhou, the soil carbon and nitrogen contents showed a unimodal trend (that is, there is only a single highest point of 1,500 m a.s.l.) with increasing altitude (Figs. 3A–3C), whereas the that of phosphorus content showed no obvious trend (Figs. 3D & 3E). This is consistent with the research results in karst mountain ecosystems (*Liu et al., 2022a*; *Liu et al., 2022b*) and other ecosystems (*Guo et al., 2022*; *Hou et al., 2019*). Previous studies have shown that there is an obvious covariant relationship between soil carbon and nitrogen (*Tashi et al., 2016*; *Yin, Wang & Zhou, 2022*). As the surface runoff in karst areas is small, the scouring and leaching effect of runoff on the topsoil is weakened, and hance, the storage of soil carbon and nitrogen is determined by the balance between carbon and nitrogen inputs from net primary productivity and outputs through microbial decomposition (*Schlesinger, 1990*).

As hypothesized by *Reich & Oleksyn (2004)*, low temperatures slow down biogeo-chemical cycles (*e.g.*, carbon cycle). As altitude increases, soil temperature decreases, water content increases, microbial and soil animal activity decreases, decomposition of apoplastic matter decreases, thereby weakening the mineralization of organic carbon and nitrogen

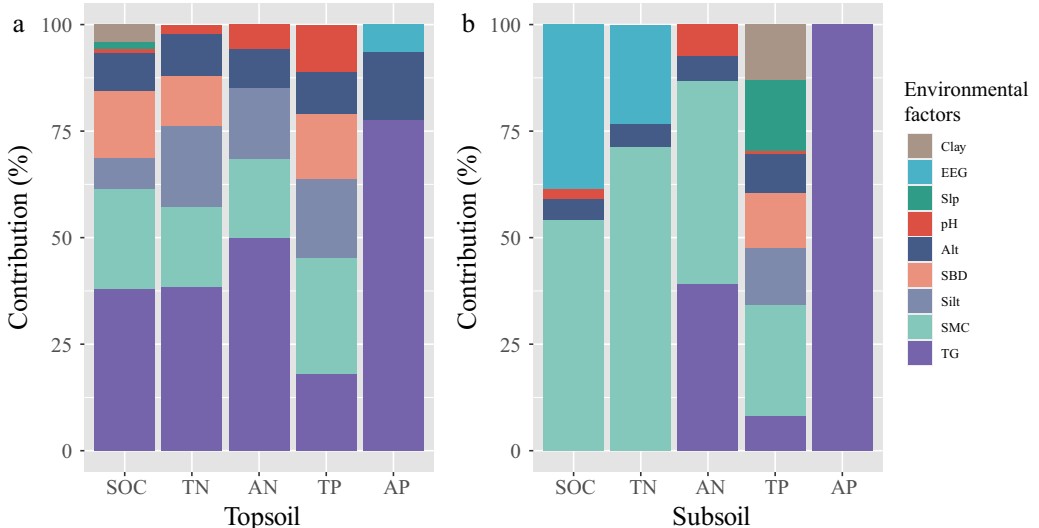

**Figure 5 The contribution rate of environmental factors to the explained variation in SCNPC in the topsoil and subsoil.** SOC, soil organic carbon; TN, soil total nitrogen content; AN, soil available nitrogen content; TP, soil total phosphorus content; AP, soil available nitrogen content; Alt, altitude; Slp, slope; SBD, soil bulk density; SMC, soil moisture content; Clay, percentages of soil clay (< 0.002 mm); Silt, percentages of soil silt (0.002–0.05 mm); EEG, easily extractable glomalin-related soil protein; TG, total glomalin-related soil protein.

and increasing the input of organic compounds, thus increasing SOC (*Bangroo, Najar & Rasool, 2017*; *Reich & Oleksyn, 2004*). However, this assumption is based on the fact that the input of net primary productivity is essentially the same at all altitudes. Therefore, the reason for the unimodal trend of soil nutrients along an altitudinal gradient may be mainly caused by altitude changes and differences in organic matter inputs. At 1,200–1,500 m a.s.l., the forest development time (approximately 50 a) and stand age were the same, and the net primary productivity inputs to soils were the same, and hance, the low apoplastic decomposition rate and low soil mineralization based in altitude eventually made the inputs of carbon and nitrogen increase with altitude. At 1,600 m a.s.l., the growth of *P. massoniana* may be affected by the lower average annual temperature and winter frost at the top of the mountain, and there is an obvious upper altitude limit (*Zhou, 2001*; *Qi et al., 2016*) and the development time of the forest community is approximately 10 a, and hance, the primary productivity is significantly lower than that of the forest at 1,200–1,500 m a.s.l., and the input of organic matter such as apoplankton is extremely low. Therefore, the soil carbon and nitrogen of the forest community decreased abruptly when the altitude increased to 1,600 m a.s.l.. Therefore, the soil carbon and nitrogen in the karst mountain *P. massoniana* forest showed a trend of increasing first and then decreasing. In contrast to carbon and nitrogen, which are mainly derived from plant photosynthetic fixation (*Samanta et al., 2011*) and atmospheric deposition (*Galloway et al., 2004*), respectively, the phosphorus in ecosystems is mainly derived from rock weathering (*Walker & Syers, 1976*). Therefore, compared with carbon and nitrogen, the phosphorus cycle in the pedosphere and biosphere is a relatively closed and slow process (*Wu et al., 2013*), which may explain

why the phosphorus cycle in the mountain forest ecosystem had no obvious law related to the variation in altitude.

## Factors influencing SCNPC in the *P. massoniana* forest

Although the effects of silt, SBD, and Alt cannot be ignored, TG (*Li et al., 2022*; *Percivall et al., 2022*) and SMC (*McLauchlan, 2006*; *Qian, Pan & Sun, 2013*; *Sheng et al., 2021*) may be the main environmental factors influencing SCNPC of topsoil in the karst *P. massoniana* forest. Our study showed that, in the topsoil, the cumulative explanatory rates of TG, SWC, silt, SBD and Alt to the explainable part of SOC, TN, AN, TP and AP were all greater than 88.72%, and the cumulative explanatory rates of TG and SMC reached 45.28–77.33%, which were consistent with the results of existing studies (*Cai, 2021*; *Zhang et al., 2019a*; *Zhang et al., 2019b*). As a metabolite of arbuscular mycorrhizal fungi, TG is an indirect way for plant and microbial communities to affect soil properties (*Li et al., 2015*), and it is beneficial for soil aggregation, quality improvement, and SCNPC storage (*Zhang et al., 2023*; *Li et al., 2015*), which are the main influencing factors effecting SCNPC (*Zhong et al., 2017*). Although *P. massnoiana* is a typical ectomycorrhizal fungal host, a large number of arbuscular mycorrhizal host plants coexist in *P. massoniana* forest (*Zhang et al., 2021*); therefore, TG has a non-negligible importent effect on SCNPC in topsoil of *P. massoniana* forest at different altitudes. Soil moisture is one of the main influencing factors of hydrologic process and plant growth and recovery, it is also the carrier of substance circulation in the soil system (*Liu et al., 2021*; *Laio et al., 2001*). In this study, there was a highly significant correlation between SMC and SCNPC, and it is the main factor affecting SCNPC, which is consistent with the results of existing studies in karst areas (*Cai, 2021*). It has been shown that soil moisture and SCNPC have an obvious coupling relationship, and soil moisture may affect the nutrient restitution process of plants and microorganisms and the mineralization and nitrification rates of carbon and nitrogen, which in turn have important effects on SCNPC (*Liu et al., 2010*; *O'Brien et al., 2010*; *Otieno et al., 2010*). In addition, although the study area receives more than 1,100 mm of rainfall throughout the year, short-term drought often occurs in the karst areas of the region during the forest growing season (*You et al., 2022*), and SMC exerts a critical limiting influence on the process of elemental biogeochemical cycling in karst forest ecosystems at this time (*He et al., 2021a*; *He et al., 2021b*). Meanwhile, the total soil volume in the study area is small, and the soil moisture holding capacity is low, and the spatial and temporal variability of water is high, the influence of SMC should not be neglected. So, soil moisture is the main factor affecting the vertical variation of SCNPC of topsoil in karst *P. massoniana* forest. Silt and SBD are important physical properties that affect the transformation and effectiveness of soil nutrients such as soil carbon, nitrogen, and phosphorus (*Hassink, 1997*; *Zhang et al., 2015*). SBD and Silt may affect SOC, TN, AN, and TP in topsoil by affecting the soil moisture (*Zhang et al., 2016*). Finally, altitude is an important factor influencing SCNPC in karst *P. massoniana* forest because it can alter microclimatic conditions such as temperature and precipitation (*Shamsher et al., 2019*), which in turn affects soil physical properties, plant and microbial community structure (*Pan et al., 2009*; *Rezaei & Gilkes, 2005*) and ultimately SCNPC (*Zhao et al., 2015*; *Siles et al., 2017*).

In the subsoil, the environmental factors affecting SCNPC are different. SOC and TN were mainly influenced by SMC and EEG, AN was mainly influenced by SMC and TG, TP was influenced by several environmental factors, and AP was only influenced by TG. As with the topsoil, SMC is still the main factor influencing SOC, TN, AN and TP in the subsoil. However, compared with the topsoil, the main sources of nutrients in the lower soil are the leaching of nutrients from the upper layer and some plant root-microbial activities (*Liu, Wang & Dai, 2019*; *Zhao, Zhou & Yan, 2010*), so the subsoil is still influenced by soil moisture. Meanwhile, EEG is also the main factor affecting SOC and TN in the subsoil, probably because the distribution of AMF roots is much less in the subsoil and the accumulation and turnover of GRSP is much lower than that in the topsoil, so the newly produced GRSP (*i.e.,* EEG) has more influence on soil carbon and nitrogen content.

In addition, many studies have shown that a variety of organic acids produced during the decomposition of *P. massoniana* apoplast can increase the acidity of the soil (*Tong & Ding, 2012*). Meanwhile, the decrease in pH facilitates the dissolution of insoluble Ca-P in the soil, thus increasing the AP (*Perez, Smyth & Israel, 2007*). However, our study found that soil pH and Ca were not significantly correlated with AP and stepwise regression analysis did not reveal significant effects of pH and Ca on SCNPC, suggesting that inorganic phosphorus in the form of Ca-P may not be the main source of AP in Sargassum pine soils at different altitudes of karst mountains (*Bai, Zhang & Wang, 2001*).

## CONCLUSIONS

This study revealed significant differences in the SCNPC (SOC, TN, AN, and TP) of the *P. massoniana* forest along an altitudinal gradient in the karst mountains of Southwest China. SOC, TN, and AN showed a significant unimodal model along the altitudinal gradient, whereas TP and AP showed no defined trends. TG and SMC may be the main factors affecting the SCNPC of the *P. massoniana* forest topsoil at different altitudes, while the main influencing factors of each environmental factor of SCNPC in the subsoil are significantly different. These findings provide a reference for a better understanding of the changes in soil properties at different altitudes in karst mountain forest ecosystems, along with their influencing mechanisms. It is necessary to strengthen research on the feedback mechanisms of vegetation–soil interactions under altitude gradients to better understand the altitudinal patterns of soil nutrients in subtropical montane forest ecosystems and understand their influencing mechanisms.

## ACKNOWLEDGEMENTS

We wish to thank the editor and anonymous reviewers for their constructive comments and suggestions for improving this manuscript.

### Funding

This research was financially funded by the National Natural Science Foundation of China (31660150, 31960234). The funders had no role in study design, data collection and analysis, decision to publish, or preparation of the manuscript.

### Grant Disclosures

The following grant information was disclosed by the authors:
National Natural Science Foundation of China: 31660150, 31960234.

### Competing Interests

The authors declare there are no competing interests.

### Author Contributions

- Kun Nie conceived and designed the experiments, performed the experiments, analyzed the data, prepared figures and/or tables, authored or reviewed drafts of the article, and approved the final draft.
- Ming Xu conceived and designed the experiments, performed the experiments, analyzed the data, authored or reviewed drafts of the article, and approved the final draft.
- Jian Zhang conceived and designed the experiments, performed the experiments, analyzed the data, authored or reviewed drafts of the article, and approved the final draft.

### Data Availability

The raw measurements are available in the Supplementary File.

### Supplemental Information

Supplemental information for this article can be found online at http://dx.doi.org/10.7717/peerj.15198#supplemental-information.

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
