# Peer review of "Changes in soil carbon, nitrogen, and phosphorus in Pinus massoniana forest along altitudinal gradients of subtropical karst mountains"

_PeerJ, doi:10.7717/peerj.15198_

## Round 0.1 · original submission · Major Revisions

The authors should highlight the contribution of the study. All the comments made by the reviewers are important and the authors need to revise the research article accordingly. It requires major revision.

Reviewer 1 ·

Basic reporting

• Although Introduction is sufficient and nicely written but establishing the interrelation climatic parameters & soil properties (pH) with carbon and nutrient dynamics may increase the connectivity and understanding.
• P. massoniana as written in heading (line 137, 147, 157, 169, 189) should be italic.
• Also explain the significance of size of circle and squares in the title of figure-4, if any.
• Match the styling of reference at line number 231, 240, 252, 257 etc. as the journal name should be in lower case which has been followed in other references.

Experimental design

• As the experimental area fall in the altitude of ~800 to 1800 m, then what can be the reason behind sampling from 1200-1600 M only?

Validity of the findings

Findings of present investigation are presented nicely using qualitative & informative graph, figures and tables as well as supported very well in the discussion with scientific avidences.
• As the Pinus leaves are known to cause soil acidity which can again to shift in nutrient dynamics specially the phosphorous availability. • Emphasize on such possible interactions as well if any was observed!

·

Basic reporting

The manuscript entitled “Changes in soil carbon, nitrogen, and phosphorus in Pinus massoniana forest along altitudinal gradient of subtropical karst mountains” has been well prepared. Professional English has been used throughout. The manuscript has been written in a lucid and very comprehensible way.

Experimental design

The experimental design is sound. The methods adopted is clearly described and can be replicated.
The research question is defined, but clarity is lacking. I made some critical observations about the manuscript and placed them before the authors for their answers/justification.
1. The authors have used HCl-HClO4 to determine available phosphorous content (ISSCAS, 1978) at line no. 104-105; however, Olsen’s/Bray’s method for neutral/acidic soil is a well-established method for estimating available P in soil. Can you please explain why you didn’t use this method?
2. Can you please let us know whether the soil was pre-treated to remove organic carbon and calcium carbonate before dispersing it as described in line no. 109-110?
3. It is well known that Karst mountains are rich in calcium, so its content should have been estimated as it would be affecting your phosphate availability in soil.
4. What about the erosional loss of topsoil? An average of 1100 mm of rainfall is received, and the study has been made at an altitude from 1200 m to 1600 m with an increment of 100 m. So, will it not affect your nutrient status?
5. You have obtained silt as one of the soil’s physical properties significantly affecting SCNPC. However, clay is the most active component. How do you explain this?
6. The data for pH is lacking. It can’t be seen anywhere in the graphs or figures. Please include it in the main manuscript.

Validity of the findings

no comment

Reviewer 3 ·

Basic reporting

1. The articles read well.
2. The literature and analysis seem adequate.
3. The introduction, results, and discussion need improvement.

Experimental design

1. The introduction failed to introduce the necessity, broad perspective, and implication of this study.
2. The results section can be improved.
3. The discussion needs major improvement. One of the primary objectives of the study was to determine the effect of altitude on SCNPC variation, but the discussion hardly touched on this topic. for example, figure 2 and 3 shows unimodal trend of SOC distribution with highest peak around 1500 m altitude, but discussion has no explanation on the possible reason for this. The discussion has no reference to or explanation of the different contributing factors to the availability and distribution of SCNPC as shown as/in Figure 5.

Authors can find the specific comments in the attached pdf file.

Validity of the findings

1. Authors need to improve the introduction section to define the rationale of the study, why it is important, and how the current results can help in future research or management ; especially for soil C management (considering future climate mitigation goals), and how the forest system can be better managed for the same.

Annotated reviews are not available for download in order to protect the identity of reviewers who chose to remain anonymous.

---

## Round 0.2 · accepted · Accept

The paper has been accepted for publication.

Reviewer 1 ·

Basic reporting

Abstract and Introduction part is qdequate and written nicely.
Issues raised have been addressed.

Experimental design

NA

Validity of the findings

Revised results and discussion are very much improved and adequate.

·

Basic reporting

No Comment

Experimental design

The authors have incorporated the suggestions in the manuscript.

Validity of the findings

The authors have incorporated the suggestions in the manuscript.

Additional comments

I am satisfied with your explanation of my queries. Furthermore, the corrections have been incorporated into the manuscript as suggested. I would recommend the acceptance of the manuscript.